# Microorganismal Cues Involved in Host-Location in Asilidae Parasitoids

**DOI:** 10.3390/biology11010129

**Published:** 2022-01-13

**Authors:** Marcela K. Castelo, José E. Crespo

**Affiliations:** Laboratorio de Entomología Experimental—Grupo de Investigación en Ecofisiología de Parasitoides y Otros Insectos (GIEP), Departamento de Ecología, Genética y Evolución, Facultad de Ciencias Exactas y Naturales, Instituto IEGEBA (CONICET-UBA), Universidad de Buenos Aires, Buenos Aires C1428EHA, Argentina; crespo@ege.fcen.uba.ar

**Keywords:** microorganismal cues, endosymbionts, host-parasitoid systems, *Mallophora*

## Abstract

**Simple Summary:**

This research shows how symbionts mediate orientation to hosts in a dipteran parasitoid system. We show that two close related species use cues located in different parts of the host. This study increases our knowledge of *Mallophora* species and sheds light on adaptations they have when adopting a parasitoid lifestyle.

**Abstract:**

Parasitoids are organisms that kill their host before completing their development. Typical parasitoids belong to Hymenoptera, whose females search for the hosts. But some atypical Diptera parasitoids also have searching larvae that must orientate toward, encounter, and accept hosts, through cues with different levels of detectability. In this work, the chemical cues involved in the detection of the host by parasitoid larvae of the genus *Mallophora* are shown with a behavioral approach. Through olfactometry assays, we show that two species of *Mallophora* orient to different host species and that chemical cues are produced by microorganisms. We also show that treating potential hosts with antibiotics reduces attractiveness on *M. ruficauda* but not to *M. bigoti* suggesting that endosymbiotic bacteria responsible for the host cues production should be located in different parts of the host. In fact, we were able to show that *M. bigoti* is attracted to frass from the most common host. Additionally, we evaluated host orientation under a context of interspecific competence and found that both parasitoid species orient to *Cyclocephaala signaticollis* showing that host competition could occur in the field. Our work shows how microorganisms mediate orientation to hosts but differences in their activity or location in the host result in differences in the attractiveness of different cues. We show for the first time that *M. bigoti* behaves similar to *M. ruficauda* extending and reinforcing that all *Mallophora* species have adopted a parasitoid lifestyle.

## 1. Introduction

Microorganisms are in intimate association with insects having negative, neutral, or positive effects on the host [1]. Given that most insect cells are microbial, the insect could be considered a multi-organism or holobiont [2]. There are many different microbial associations with the host, and they can be found in different parts of the insect body. Microorganisms can be found on the insect cuticle, gut, or even inside cells [2]. The cuticle is often the habitat for many microorganisms but with a limited capacity of establishing due to different factors like physical disturbance or antimicrobial secretions [2]. In turn, the gut can host many microorganisms and even many insects have specialized structures to harbor them, i.e., the ileum in the scarab beetles [2]. This structure, named the fermentation chamber, harbors symbiotic bacteria that aid in digestion by breaking down cellulose [3,4]. Finally, many specialized insect cells can house microorganisms like bacteriocytes or mycetocytes [2,5]. Microorganisms have multiple effects on their hosts and play different roles. There are plenty of examples that microorganisms have important functions on nutrition, protection against natural enemies, detoxification of toxins, host location, and in intra and interspecific communication serving as sources of cues and signals [2,6,7,8,9]. Interestingly, microorganisms have been postulated as responsible for the synthesis of insect semiochemicals that could modify the behavior or activities of many insects [10,11].

For parasitoids, chemical cues related to the host can provide reliable information indicating the precise location of the host [12,13,14]. These cues can be originated from the host body themselves, associated microorganisms or even from direct host activities as oviposition or feeding where plant chemicals in response to herbivory or oviposition injuries, host salivary secretions, or host frass are between the most important [12,14,15]. Among dipteran parasitoids, many different host location strategies can be found compared to the hymenopteran parasitoids where the female locates the host and places its eggs [12,14,15,16]. The most widely dipteran parasitoids group studied are the Tachinidae that are all endoparasitoids with diverse host location strategies. Some species place their eggs directly on the hosts, others have larvae with active host searching behaviors while some species have a waiting type strategy by placing the egg on leaves that must be eaten by hosts [13,15]. Regarding cues, some Tachinidae species are attracted to damaged plants while others can be attracted to host secretions or frass [13,15]. Notwithstanding there is some information on tachinid species, other groups have been far less studied and little information is available.

The vast majority of the information available of the strategies and cues involved in host location deals with above-ground hosts but far less is known for soil-dwelling hosts. Underground habitats have several characteristics like pore size, moisture, or gas gradients that can serve as confounding factors during host location [17,18]. Scarab beetle larvae are important members of soil fauna. They are mainly considered pests since they are seed and root feeders that can damage several agricultural crops [19]. They are voracious root feeders that feed on cellulose and hemicellulose aided with symbionts present in a specialized fermentation chamber in the hindgut [20]. Given their importance as pests, several techniques to reduce their population have been tested [18]. Besides chemical pesticides, biological control through parasitoids has been suggested as a possible strategy [19]. The most frequent parasitoids mentioned in the literature as scarab beetle enemies are *Tiphia* wasps and many Tachinidae flies [19].

In this work, we will introduce two dipteran parasitoid species belonging to the Asilidae. This family is well known because adults, commonly named robber-flies, are highly voracious predators of other insects [21]. Although almost every species known has predatory larvae, two species are belonging to the *Mallophora* genera with a documented parasitoid lifestyle [22].

*Mallophora ruficauda* Wiedemann 1828 (Diptera: Asilidae) is an endemic robber fly of Argentina, particularly of the Pampas region inhabiting open grasslands near bee farms [23]. Adults predate on insects such as bees, other hymenopterans and dipterans while larvae are solitary koinobiont ectoparasitoids of scarab beetle larvae, i.e., soil white grubs (Coleoptera: Scarabaeidae) (Figure 1D). *Mallophora ruficauda* has a high preference for parasitizing both second and third instar larvae of the Argentinian scarab, *Cyclocephala signaticollis* Burmeister 1847 (Coleoptera: Scarabaeidae) in the field [23]. Nine species have been recorded for the Pampas region: *C. signaticollis, C. putrida* Burmeister 1847, *C. modesta* Burmeister 1847, *Diloboderus abderus* Sturm 1826, *Arcophileurus vervex* Burmeister 1847, *Anomala testaceipennis* Blanchard 1856, *Bothynus striatellus* Fairmaire 1878, *Heterogeniates bonariensis* Ohaus 1909, and *Plectris bonariensis* Bruch 1909 (=*Philochloenia bonariensis* Bruch 1909) [24].

*Mallophora ruficauda* females place their eggs on tall grasses away from underground hosts (Figure 1A). Parasitoid larvae are dispersed by the wind, they bury themselves once they reach the soil and, after molting to the second instar, actively locate and parasitize hosts (Figure 1B). Orientation to the host is mediated by chemical cues arising from microorganisms inhabiting the hosts’ fermentation chamber but the identity of the symbionts generating them is unknown [25]. It is thought that these substances are involved in the host chemical communication system that parasitoid larvae can exploit [26]. Previous information has shown that *M. ruficauda* larvae can assess host instar and parasitism status of the host through these chemical cues [23]. Orientation to the host is modulated by the preparasitism competition inducing the attractiveness of suboptimal hosts [23].

There exists another species, *M. bigoti* Lynch Arribálzaga 1833 that is found frequently associated with beehives [27] (Figure 1C). This species can outnumber *M. ruficauda* in some areas leading to high predation pressure on honeybees (J. Crespo and M. Castelo, personal observations). However, ecology and life history studies are completely absent for this species. Presumably, as *M. bigoti* belongs to the *Mallophora* genus, it is also a white grub ectoparasitoid but it is unknown which species does it attack if it has selectivity for some host species, how it finds the hosts, and where it places its eggs.

The population of both species seems to have increased in the last few years given the increasing damage reports from beekeepers. In Argentina, the zones with the greatest damage from *M. ruficauda* is the Pampas region comprised of Buenos Aires, Entre Ríos, Santa Fe, Córdoba, and La Pampa provinces where it has been estimated that economic losses by robber-flies are around 2 and 3 million dollars annually [28]. This region is among the most productive of Argentina and concentrates about 87% of the beehives of the country [29]. The main damage that *M. ruficauda* produces is through the indirect interference with the agriculture activities in crops that involve insect pollination resulting in a clear reduction of fruits and seeds [30]. This is because *M. ruficauda* preys on worker bees when they are flying to collect nectar and pollen from different flowers. The importance of *M. bigoti* as a honeybee predator has yet to be established, but it has been seen that it has also the capacity to prey on honeybees adding to the total damage the *M. ruficauda* can inflict.

Hence, given the importance of these predators of honeybees and the fact that much information is still missing on the ways that parasitoid larvae find the host we studied: (a) if scarab larval endosymbionts are responsible for the generation of the cues involved in host-orientation in *M. bigoti*, (b) whether *M. bigoti* has specificity for a particular species, and (c) the effect that interspecific interaction during host-location has on orientation in these two species.

## 2. Materials and Methods

### 2.1. Insects

Larvae of both *M. ruficauda* and *M. bigoti* were collected from egg-clusters during January 2019 to March 2021 on grasslands near bee farms in Castelli (36°04′ S, 57°49′ W), Mercedes (34°37′ S, 59°27′ W), Moreno (34°46′ S, 58°93′ W), Pilar (34°28′ S, 58°55′ W), and Sierra de los Padres (37°55′ S, 57°40′ W), localities associated with apiaries in Buenos Aires province, Argentina (Figure 2). In the field, egg-clusters were carefully cut off from their support and were kept individually in Falcon-type tubes until the larvae hatched. After hatching, the neonate larvae were kept in flasks (diameter = 6.8 cm; height = 12.3 cm), containing 100 mL of potting soil as substrate at a density of 3 larvae/mL of soil. Each flask contained larvae of several egg-clusters. Larvae were stored in darkness and at a controlled room temperature of 25 ± 1 °C. When larvae reached the second instar and were 20 to 50 days old, they were used to perform the experiments. Age is an important factor since it has been already established for *M. ruficauda* that recently molted larvae do not orientate to the host [31].

Scarab larvae were collected at a soil depth of 0.30 m in grasslands of Castelli, Mercedes, Moreno, Pilar, and Sierra de los Padres in Buenos Aires province, Argentina, from February 2019 to March 2021. A random sampling of the soil was performed near the apiaries using a common shovel picking every white grub found. Collected white grubs were taken to the laboratory and identified using a taxonomic key [32]. Only third instar larvae of *C. signaticollis*, *C. modesta*, *C. putrida*, *P. bonariensis*, and *H. bonariensis* were used since they were the most abundant species found during sampling. Host larvae were maintained individually in the laboratory at a room-controlled temperature at 25 ± 1 °C in black tubes of 30 mL filled with potting soil and fed weekly with pieces of fresh carrots.

### 2.2. Experimental Procedures

#### Host Orientation

To evaluate the orientation response to the hosts we performed dual choice experiments in a static air two-way olfactometer (Figure 3). The olfactometer consisted of a rectangular plastic box of general dimensions of 24 cm by 15 cm and 4 cm tall. Inside, the box was divided into six small arenas of 15 cm by 4 cm and 4 cm that were the individual chambers where only one parasitoid larva was tested per chamber. Each individual chamber was divided into three equal-size zones (one middle and two laterals) along the long axis (see [23,31]). As a stimulus, we used live hosts that were placed on one of the lateral zones and a plastic mesh that allowed the parasitoid larva to move freely around the arena but kept the host on the lateral zone. For the control series, no live hosts were placed in the arena. This series allowed us to detect any possible asymmetry effect inherent to the experimental device.

In each assay, only one larva from a given species was gently released with a paintbrush at the center of the arena. Both hosts and parasitoids were released simultaneously. Then, the box was kept under complete darkness with a piece of humid filter paper between the box cover and the box to keep the relative humidity high inside the experimental arena. After 60 min, the position of the larva was registered. Three possible responses were scored according to the position of the larva in one of the three zones of the arena: choice for the stimulus, choice for the empty zone or control, or no decision if the larva was found in the middle zone (Figure 3). The larvae that were found in the middle zone were excluded from the analysis. After every trial, each individual was removed and the arena was cleaned up with soap, water, ethyl alcohol and then dried with an air current to eliminate any possible remaining cue. All experiments were conducted between 8:00 and 18:00 h on days where the barometric pressure was stable or increasing because it has been shown that drops in barometric pressure halt the orientation behavior of the larvae [33]. Experiments were carried out under laboratory conditions (25 ± 1.0 °C) and in darkness. Each individual, either larva or host was used only once in the experiment. A double control series was performed where no stimulus was offered, and we studied the spontaneous orientation activity of larvae in the experimental arena. This series allowed us to test if there existed any asymmetry in the experimental conditions that could introduce any bias in our interpretation of results.

### 2.3. Symbiont Based Cues and Host Specificity during Host Orientation

The rationale behind this experiment was to study if endosymbionts from the fermentation chamber are responsible for the production of the chemical cues involved in host orientation. To this, we treated soil with tetracycline at 0.1%*w/w* concentration and use it as the feeding substrate for host larvae of the most abundant species that were found at the collection sites of parasitoid larvae for seven days in the black tubes (30 mL) with pieces of carrot. As control, we used untreated hosts of the same species maintained in the same way. After seven days, we used host species as a stimulus for host orientation experiments as explained above. Additionally, we further tested the orientation of *M. bigoti* to host frass from species where positive orientation was obtained. To this, we collected fresh frass pellets from different species hosts and weighed them to ensure that amounts of offered frass were similar between experiments. We then placed the frass individually on pieces of filter paper to be used as the stimulus. We then tested orientation in the same manner as before. Treatments and replicates are detailed in Table 1.

### 2.4. Interspecific Competence during Host Orientation

Given that *M. ruficauda* is the most important species that attacks honeybees and that *C. signaticollis* is its preferred host, we studied if the orientation to the host was modified if the host was already parasitized by the other species, *M. bigoti*. To this, we tested the orientation of either a larva of *M. bigoti* or *M. ruficauda* to previously parasitized hosts of either a conspecific or heterospecific larva. Hosts were parasitized between 7 and 10 days before because we have already shown that less time is not sufficient to change the chemical identity of the host. As a control series, we tested orientation in larvae of both species to healthy hosts. Treatments and replicates are detailed in Table 2.

### 2.5. Statistical Analyses

To model the probability of orienting to a host we used a Binomial GLMM with a logit link function for each parasitoid species. The logit link function ensures fitted values between 0 and 1, and the Binomial distribution is typically used for probability data. Fixed factors were host species (categorical with three levels) and host treatment for hosts either treated with tetracycline or not (categorical with two levels). To incorporate the dependency among observations of the same year, egg-cluster, and olfactometer (recall each olfactometer consisted of six arenas), we used year (only for the *M. ruficauda* model because all experiments for *M. bigoti* were performed on the same year), egg-cluster, and olfactometer as random factors. Every time the parasitoid larva oriented to a host was counted as a success (noted with a 1), otherwise, a 0 was registered. Hence, this model included both fixed factors with its interaction (host species by treatment) and three random factors.

Then, to model the probability of *M. bigoti* of orienting to host frass, we also used a Binomial GLMM with a logit link function. The fixed factor was host species (categorical with two levels). To incorporate the dependency among observations of the same egg-cluster and olfactometer, we used egg-cluster and olfactometer as random factors. Every time the parasitoid larva oriented to a host was counted as a success (noted with a 1), otherwise, a 0 was registered.

Finally, to model the probability of *M. ruficauda* or *M. bigoti* of orienting to host parasitized by conspecifics of heterospecifics, we also used a Binomial GLMM with a logit link function. Fixed factors were searching host species (categorical with two levels, *M. ruficauda* or *M. bigoti*) and host status (healthy, parasitized with *M. ruficauda* or with *M. bigoti*). To incorporate the dependency among observations of the same egg-cluster and olfactometer, we used egg-cluster and olfactometer as random factors. Every time the parasitoid larva oriented to a host was counted as a success (noted with a 1), otherwise, a 0 was registered. Hence, this model included both fixed factors with its interaction (searching host species by host status) and three random factors. Control series with no stimulus were analyzed through Chi-Square goodness of fit tests to assess any bias from our experimental device into the experiments.

All the analyses were done using the R v3.6.3 “Holding the Windsock” software [34]. The package glmmTMB and nlme were used to fit the models [35,36]. For testing model assumptions, we used the package DHARMa [37]. Graphs were done using the package ggplot2 [38]. Tukey contrasts were performed with the emmeans function of the package emmeans [39].

## 3. Results

### 3.1. Symbiont Based Cues and Host Specificity during Host Orientation

Our control experiments in which we tested for the adequacy of the experimental device to study orientation revealed no difference for any particular side of the arena for both parasitoid species (χ-squared = 0.0579, df = 1, *p*-value = 0.8099). Regarding the percentage of replicates of trials excluded due to larvae found in the middle zone, we excluded 32.58% (N = 84) of the replicates for *M. ruficauda* and 43.85% (N = 139) of the replicates for *M. bigoti*.

We found that both *Mallophora* species can orientate to the host through chemical cues arising from the host (Table 3).

Regarding *M. ruficauda*, we had already established that this species uses the chemical cues present in the fermentation chamber of *C. signaticollis* during host orientation [24,26]. We found a 0.827 probability of orienting to a healthy host (Table 4). However, when *C. signaticollis* was pretreated with tetracycline we found a probability of 0.505 of orienting to the host (Table 4). Additionally, we found the same orientation patterns to the close species *C. putrida* but not to *C. modesta* showing that *M. ruficauda* does not orientate to this host species (Table 4).

The results for *M. bigoti* were substantially different from those found for *M. ruficauda*. Host orientation experiments evinced that *M. bigoti* orientates to *C. signaticollis* and *H. bonariensis* host species but not to *P. bonariensis* (Table 4). Interestingly, treatment with tetracycline did not modify any of the orientation patterns for any of the three species (Table 4).

Given that tetracycline did not modify the orientation patterns of *M. bigoti* to hosts we performed host orientation experiments but offered host frass of *C. signaticollis* and *H. bonariensis* as stimulus to *M. bigoti*. Experiments revealed that *C. signaticollis* frass is not attractive to *M. bigoti* (Probability: 0.606 (LCI: 0.429, UCI: 0.759); *p* = 0.2322). On the contrary, *H. bonariensis* frass was attractive to *M. bigoti* (Probability: 0.750 (LCI: 0.515, UCI: 0.894); *p* = 0.0382).

### 3.2. Interspecific Competence during Host Orientation

We found that both *Mallophora* species orientate to *C. signaticollis* regardless of the host status. In both species, if the host was healthy or parasitized (with a conspecific or heterospecific), the parasitoid larva was attracted to the host (Table 5).

## 4. Discussion

In this work, we studied if host orientation is mediated by chemical cues originating from the host in two parasitoid species of the genus *Mallophora*. This genus belongs to the Asilidae family that are thought to be the only predators. We have proven already that *M. ruficauda* has a parasitoid lifestyle as previous information suggested and propose it could be a common characteristic of all *Mallophora* species [24,31,40]. We were also able to show that another species, *M. bigoti*, has a very similar behavior regarding host orientation guided through chemical cues from the host.

Regarding *M. ruficauda*, we obtained similar results as previously published that showed that parasitoid larvae are attracted to live hosts of *C. putrida* and *C. signaticollis* but not *C. modesta*. However, when hosts were pretreated with tetracycline, orientation to *C. putrida* and *C. signaticollis* was lost. Our results show that the confidence interval for the odds ratio of *C. putrida* would indicate that there is no statistical difference, but we sustain that the marginal significance found is more prone to be because of variability that was not controlled in our experiments. Increasing the number of replicates could clarify this result in the future. Tetracycline is a widely used antibiotic that attacks a wide range of microorganisms including gram-positive and gram-negative bacteria, chlamydiae, mycoplasmas, rickettsiae, and protozoan parasites [40]. These results are in clear support for the microorganism-based host orientation. Furthermore, host orientation in *M. ruficauda* has been well established as parasitoid larvae are attracted to extracts from the fermentation chamber of *C. signaticollis* [26]. Regarding host specificity, *M. ruficauda* larvae were only attracted to *C. signaticollis* and *C. putrida*, two very related Dynastinae species but not to *C. modesta* nor to other species like *P. bonariensis* or *H. bonariensis* [31].

The results found for the other parasitoid species, *M. bigoti*, were quite different. To begin with, we found that *M. bigoti* larvae are attracted during host orientation to *C. signaticollis* and *H. bonariensis*. This species may have a more generalist diet since it is attracted to species from different subfamilies, Dynastinae and Rutelinae. When hosts were treated with tetracycline, no changes in host orientation were evident, indicating that if orientation involves microorganisms, they should be located at a different part of the host. Another alternative possibility is that since substances reaching the colon are mostly degraded, tetracycline could be less effective in preventing bacterial development leaving the host frass unaltered. Finally, it could be possible that the microorganismal community from the colon is less affected by tetracycline compared to that from the hindgut [3,4,6,18]. These results could also be indicating that orientation in *M. bigoti* is not based on microorganisms, and they may be using other sensory modalities. Orientation to host frass showed different results for *C. signaticollis* and *H. bonariensis*. Frass from *H. bonariensis* was attractive while no orientation was found for *C. signaticollis* frass. This result is of particular interest since it shows that *M. bigoti* would be using cues arising from different parts of the hosts depending on the species. In addition, to further support these results, we have performed some preliminary experiments on host acceptance and found that untreated *C. signaticollis* are highly parasitized while treated hosts are parasitized at random (unpublished data).

Finally, when evaluating if *M. bigoti* could interfere in the orientation of *M. ruficauda* to its preferred host, *C. signaticollis*, we found that orientation was not influenced by the parasitism status nor if the hosts were parasitized by conspecifics or heterospecifics. However, this result might lead to multiparasitism and competence for the host if the larva that is searching for a host is not able to recognize whether a host is already parasitized. Ultimately, it might end up in sharing the host but still leading to just one individual winning in solitary parasitoids. In addition, our results allowed us to test if *M. bigoti* avoids superparasitism. From our experiments, we can conclude that *M. bigoti* orients to the host (at least *C. signaticollis*) irrespective of its parasitism status. However, it should be noted that in these experiments, *M. bigoti* larvae were raised only with other larvae born from the same egg-cluster. In *M. ruficauda*, we have already shown that orientation to suboptimal hosts is modulated by the density of conspecifics prior to parasitism leading to larvae that are highly motivated to orientate and parasitize many hosts that are rejected if they are raised under a low density of conspecifics [23]. So, if *M. bigoti* shows a density-dependent response in orientation to suboptimal hosts, like *M. ruficauda* orientation to different hosts, it could be explained by the fact that they were raised under conditions of high density. Another possibility that has to be ruled out is if *M. bigoti* is a solitary parasitoid as *M. ruficauda* or if its host can harbor more than one parasitoid. It now becomes evident that more focused experiments controlling densities and testing on larger host parasitism contexts will aid in the understanding of the strength of both species as competitors and try to understand if *M. bigoti* could be a competitor for the host or if only marginally share a host with *M. ruficauda*.

Our results show clearly that microorganisms are an important source of information for both asilid parasitoid species during orientation to the host. However, we venture that *M. ruficauda* mainly uses the information provided by the endosymbionts located in the fermentation chamber of hosts, while *M. bigoti* probably uses information from host frass or another microorganismal source. Both sources of information are also used by other parasitoids like *Tiphia* wasps [15]. For instance, *T. vernalis* and *T. pygidialis* elicit different searching patterns when detecting host odors (*C. lurida* and *C. borealis*) or trails from feces [15]. The fermentation chamber is a very specialized part of the hindgut of the Scarabaeidae where anaerobic bacteria digest plant cellulose materials. Several methanogenic bacteria have been determined in the fermentation chamber of *Melolontha melolontha* which could be responsible for generating some attractive chemical cues [41]. Host frass is produced in the colon and rectum as a result of digestion and after the reabsorption of nutrients. The rectum is a zone with different pH and redox potential compared to the hindgut and harbors different microorganisms [4]. Given the different microorganism communities that proliferate in these two parts of the digestive tracts, it also seems possible that treatment with tetracycline acts as a more effective antibiotic on the fermentation chamber than on the colon or rectum. In fact, transit of food in the fermentation chamber can take up to four days while it takes hours on the midgut of *M. melolontha* which could explain that tetracycline acts mainly on the fermentation chamber [41].

## 5. Conclusions

In summary, our experiments suggest that *M. ruficauda* and *M. bigoti* use chemical cues from host-related microorganisms produced in different parts of the host. An interesting hypothesis emerges related to the fact that *M. ruficauda* seems to be attracted only to Dynastinae species while *M. bigoti* would be more generalist and orienting to Dynastinae and Rutelinae hosts. Further experiments with increased host species and testing for more species from different subfamilies present in the Pampas region will shed light on this interesting possibility. Finally, our work is also an important contribution to the knowledge of the biology of *M. bigoti* that, until now, had been neglected. This species is becoming increasingly more abundant, and its predatory effect on honeybees may add to that already exerted by *M. ruficauda*. Further studies on the mechanisms that larvae use for locating the host as well as insight on life-history strategies will help to aid in the monitoring and mitigating the damage that these important plagues inflict on apiculture.

## Figures and Tables

**Figure 1 biology-11-00129-f001:**
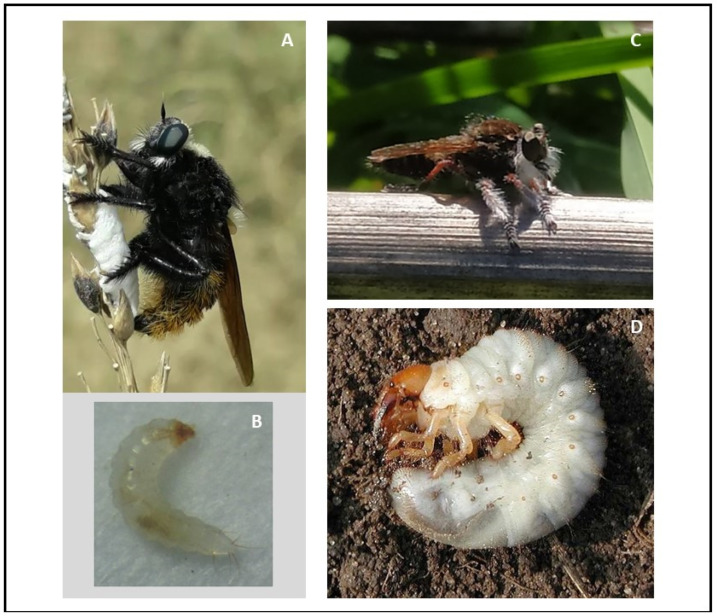
Parasitoid and host species. (**A**) *Mallophora ruficauda* female ovipositing in a plant. (**B**) Larva of second instar of *M. ruficauda*. (**C**) *Mallophora bigoti* female resting on a dry stick. (**D**) General aspect of a third instar white grub.

**Figure 2 biology-11-00129-f002:**
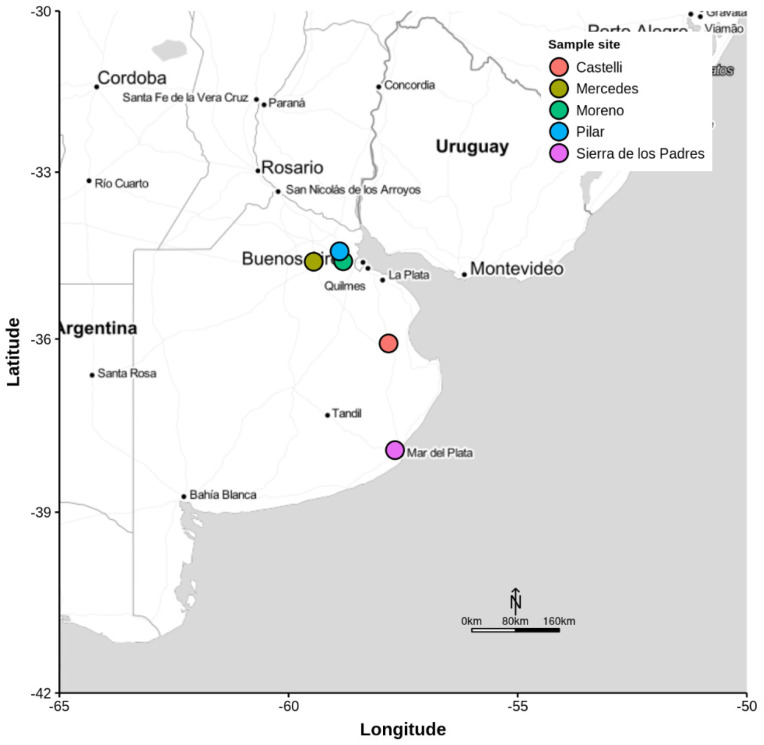
Map of the sampling sites for collecting both robber-fly egg-clusters and scarab larvae.

**Figure 3 biology-11-00129-f003:**
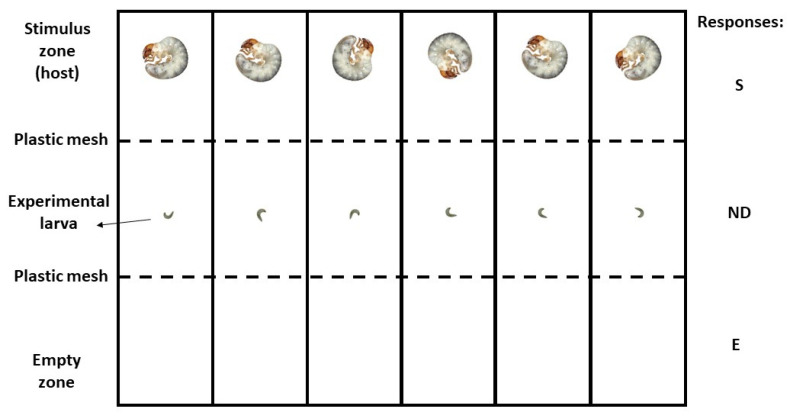
Diagram of the static air two-way olfactometer to measure the parasitoid host orientation to scarab beetle larva in dual choice tests. The device has six individual arenas. Responses: S: choice for the stimulus; E: choice for the empty zone or control; ND: no-decision. Each host zone was delimited by a plastic mesh that prevented hosts from leaving the zone.

**Table 1 biology-11-00129-t001:** Summary of the treatments for testing host orientation in *Mallophora ruficauda* and *M. bigoti*. MR: *M. ruficauda*; MB: *M. bigoti*; CS: *Cyclocephala signaticollis*; CM: *C. modesta*; CP; *C. putrida*; HB: *Heterogeniates bonariensis*; PB: *Plectris bonariensis*.

Parasitoid Species	Stimulus	Host Species	Host Treatment	N (*Host orientation)*
MR	empty			62
	Alive host	CM	untreated	30
			tetracycline	26
		CP	untreated	25
			tetracycline	30
		CS	untreated	29
			tetracycline	29
MB	empty			34
	Alive host	CS	untreated	34
			tetracycline	24
		HB	untreated	23
			tetracycline	32
		PB	untreated	23
			tetracycline	24
	Frass	CS	untreated	33
		HB	untreated	20

**Table 2 biology-11-00129-t002:** Summary of the treatments for testing host orientation in *Mallophora ruficauda* and *M. bigoti* to previously parasitized *Cyclocephala signaticollis* with conspecifics or heterospecifics. MR: *M. ruficauda*; MB: *M. bigoti*.

Searching Parasitoid Species	Host Status	N
MR	healthy	26
	parasitized with MR	24
	parasitized with MB	25
MB	healthy	29
	parasitized with MR	29
	parasitized with MB	29

**Table 3 biology-11-00129-t003:** Summary of the general results of the model for the effect of host species and treatment with tetracycline on the probability of orienting to a host for *M. ruficauda* (upper part) and *M. bigoti* (lower part). The variances of random effects are shown on the left part of the table. Results on the overall significance for fixed effects are shown on the right part of the table. Chisq: Value of the statistic. df: degrees of freedom. Pr (>Chisq): Probability that the statistic is within expected under the null hypothesis.

*Mallophora ruficauda*					
Random Effects	Variance	Fixed Effects	Chisq	df	Pr (>Chisq)
*olfactometer*	9.55^−2^	host species	2.3701	2	0.3057
*egg-cluster*	1.75^−1^	host treatment	4.3804	1	0.0364
*year*	7.42^−10^	host species × host treatment	4.1314	2	0.1267
** *Mallophora bigoti* **					
*olfactometer*	9.86^−9^	host species	2.3117	1	0.1284
*egg-cluster*	4.02^−2^	host treatment	5.6645	2	0.0589
		host species × host treatment	0.6215	2	0.7329

**Table 4 biology-11-00129-t004:** Summary of the results of the model for the effect of host species and treatment with tetracycline on the probability of orienting to a host for *M. ruficauda* and *M. bigoti*. The first part shows the predicted probabilities and odds ratios for host orientation experiments of *M. ruficauda*. The inferior part shows the predicted probabilities and odds ratios for host orientation experiments of *M. bigoti*. CM: *Cyclocephala modesta*, CP: *C. putrida*, CS: *C. signaticollis*, HB: *Heterogeniates bonariensis*, PB: *Plectris bonariensis*, df: Degrees of freedom. LCI and UCI: Lower and upper confidence interval for the estimated probability. OR (LCI;UCI): Odds ratio with lower and upper confidence interval for the odds ratio between untreated hosts/hosts treated with tetracycline. It should be noted that if the confidence interval includes 1, the OR is not significant, rendering no difference between groups.

** *Mallophora ruficauda* **								
**Host Species**	**Treatment with Tetracycline**	**Probability**	**Std. Error**	**df**	**LCI**	**UCI**	**p**	**OR (LCI;UCI)**
*CM*	no	0.534	0.112	161	0.321	0.730	0.7613	1.06 (0.298;3.78)
	yes	0.519	0.123		0.294	0.737	0.8741	
*CP*	no	0.827	0.095		0.563	0.947	0.0196	4.33 (0.847;22.13)
	yes	0.525	0.111		0.317	0.725	0.8214	
*CS*	no	0.871	0.067		0.676	0.956	0.0016	6.62 (1.536;28.48)
	yes	0.505	0.116		0.290	0.719	0.9639	
** *Mallophora bigoti* **								
**Host Species**	**Treatment with Tetracycline**	**Probability**	**Std. Error**	**df**	**LCI**	**UCI**	**p**	**OR (LCI;UCI)**
*CS*	no	0.853	0.061	153	0.690	0.938	<0.001	1.53 (0.384;6.672)
	yes	0.792	0.083		0.585	0911	0.009	
*HB*	no	0.870	0.070		0.662	0.958	0.003	2.22 (0.513;9.627)
	yes	0.750	0.077		0.573	0.870	0.008	
*PB*	no	0.609	0.102		0.401	0.783	0.303	0.78 (0.234;2.593)
	yes	0.667	0.096		0.460	0.825	0.112	

**Table 5 biology-11-00129-t005:** Summary of the results of the model for the effect of interspecific competence on the probability of orienting to *C. signaticollis* for *M. ruficauda* and *M. bigoti*. The table shows the predicted probabilities and odds ratios for host orientation experiments. df: Degrees of freedom. LCI and UCI: Lower and upper confidence interval for the estimated probability. OR (LCI; UCI): Odds ratio with a lower and upper confidence interval for the odds ratio between the species that are orienting to the host. It should be noted that if the confidence interval includes 1, the OR is not significant, rendering no difference between groups. None: corresponds to the positive control for orientation of both species to a healthy *C. signaticollis*.

Species Parasitizing Host	Species Orienting to Host	Probability	Std. Error	df	LCI	UCI	p	OR (LCI;UCI)
*M. ruficauda*	*M. bigoti*	0.759	0.080	166	0.572	0.881	0.0091	0.42 (0.11;1.63)
	*M. ruficauda*	0.882	0.055		0.724	0.955	0.0002	
*M. bigoti*	*M. bigoti*	0.793	0.075		0.608	0.905	0.0039	0.96 (0.25;3.66)
	*M. ruficauda*	0.800	0.080		0.598	0.915	0.0062	
*None*	*M. bigoti*	0.828	0.070		0.645	0.927	0.0017	1.14 (0.29;4.55)
	*M. ruficauda*	0.808	0.077		0.611	0.918	0.0044	

## Data Availability

Data available in a publicly accessible repository. The data presented in this study are openly available in FigShare at 10.6084/m9.figshare.17062373, reference number [42].

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
