# Peer review of "Microorganismal Cues Involved in Host-Location in Asilidae Parasitoids"

_biology, 2022, doi:10.3390/biology11010129_

Round 1

Reviewer 1 Report

Comments below refer to line numbers (L##) where appropriate.

INTRODUCTION

L52-54: Please clarify “direct activity”, as I am not familiar with that term. The examples given don’t seem to fit together, as only plant chemicals are an example that doesn’t fit into the categories of originated from the hosts themselves or from associated microorganisms. Salivary secretions and host frass come from the host itself, right?

L62: Should this line specify “tachinid” species instead of “dipteran” species? As written, it is confusing.

L125: It was indicated earlier (L96-97) that numerous studies have already shown that larval symbionts are responsible for orientation cues used by M. ruficauda, so it doesn’t seem appropriate to include this species in part a) of the study goals.

METHODS

L131: Were larvae of both parasitoid species collected from all five sites? Documenting the degree to which these species overlap is important in understanding competition between them.

The point above is especially important considering the results from the competition experiment. The extent of spatially, temporal, or even micro-niche (e.g., soil depth or type) overlap between the species may preclude the evolution of meaningful responses to host infection status with another species?

L147: Were soil samples taken near the apiaries?

L157-162: The design of the olfactometer is important to understanding the methods and results (e.g., understanding how far the larvae must travel to find a host). Please include a figure of the device as a diagram and/or picture, especially since the citations provided do not include such a figure.

Please clarify some details about the olfactometer. What is the box made of, plastic? Was soil present in the box during testing? How was the filter paper used, as a substrate in the box? Were the host and parasitoid placed into the olfactometer simultaneously?

L172: If the parasitoids were only checked once after 60min, can it be stated with confidence that those observed in the middle zone had not left that zone? Isn’t it possible they had left the zone and returned at the time of data collection?

L179-183: Unless there is a difference I don’t understand, this repeats information from L164-166.

L188: How was the antibiotic applied to the larvae?

Table 1: If “none” means that the host was unmanipulated, please clarify. I think a better term can be used to distinguish hosts without antibiotic treatment from the control runs in the olfactometer where there was no host present. These host-absent trials could also be called “none,” so there is potential for confusion here.

Furthermore, the number of control series (with no host present) should be indicated in Table 1 or elsewhere in the methods.

Were any individuals (parasitoids or hosts) used more than once? If not, please specify that important methodological detail. If so, please detail the extent of this and how any resulting pseudo-replication was handled.

L193-195: How much frass was used? Were the amounts similar between the two species tested in the frass assay? If so, that is interesting! If not, differences in the amount of frass could explain the differences in orientation behavior, obviating later discussion.

L204-206: How long ago were the hosts parasitized before use in this assay?

L236: Should read “discard any bias,” though I think “assess” would be a more appropriate term than “discard.”

Was the order of testing (for both hosts and parasitoids) controlled? Were randomized trials conducted, or might there by some systematic bias from testing one group before another (either over the whole course of the experiment or within each day of testing)?

Egg cluster was considered as a random effect, but how was collection site handled? Furthermore, were hosts and parasitoids always paired from the same site, or were some individuals paired with hosts from a different locality? Fine-scale local adaptation to regional hosts (even within a species) could explain some of the results obtained.

RESULTS

Please report the number (or percentage) of trials excluded because the parasitoid was found in the middle zone.

L252-253: The conclusion that the orientation response was lost upon antibiotic treatment is an interpretation of the data, better suited for the discussion section.

L253-255: It’s unclear what is meant by “orientation patterns.” For CS and CP, there was orientation towards the untreated host, but no orientation towards the untreated host. However, CS and CP were not similar in terms of the effectiveness of antibiotic in mediating the orientation response, as only the odds ratio for CS was significant. This point should be made more clearly.

L268-269: As above, this needs to be phrased to better match the statistics. The only significant orientation response was to CS, based on the odds ratios. Because of this, both parasitoid species have comparable responses: significant orientation towards CS but no orientation towards other (not shared) species.

L271: Save the interpretation of the data (i.e., use of different orientation cues) for the discussion.

Table 3. What are the 2nd/3rd indicators in the table? The methods (L149-151) indicated only third instar hosts and second instar parasitoids were used. I expected the table to be formatted with merged cells for each host, just as was done for CM, HB, and PB.

Table 3. The UCI for PB has two decimal points and is missing the closed parenthesis.

Figure 2. The bar graph display is not very informative for these data. This would be more effectively displayed as a box plot with a reference line at 0.5 probability. Then, confidence intervals overlapping 0.5 would indicate no orientation response in that treatment. Confidence intervals below 0.5 would indicate avoidance, above would indicate attraction.

L282: The reference should be for Table 4.

Table 4 formatting is odd, raising the same questions I had about Table 3.

DISCUSSION

L302-303: According to the odds ratio, antibiotic did not affect the orientation to CP.

L316-317: Is there any evidence of antibiotic having location-specific effects? It seems far more likely that cues of another modality are used, though the frass results are intriguing.

Have tests been done to test the efficacy of the antibiotic? In cases where there do not appear to be antibiotic effects, it could be that the initial dose was not sufficient to clear the symbionts from the host. Alternatively, re-infection from the environment (e.g., carrots or soil) could help explain these results.

L321-322: Given the uncertainties about antibiotic efficacy, it seems premature to conclude anything about the different parts of the host creating cues. This conclusion reaches beyond the data and seems less plausible than other explanations.

The arguments about parasitoid density (L331-344) are compelling, but another highly plausible alternative should be considered: the parasitoid may still orient toward an infected host, but it may then reject that host after detecting more information. The studies presented here only represent an early step in the parasitism process: locating a host. After location, the parasitoid would have further opportunities to reject the previously infected host and seek another.

L345-346: The argument for a reliance on chemical cues is compelling, but not complete. Have vibratory cues from the hosts been ruled out as a potential signal for seeking parasitoids? Vibrations are an important source of information in the darkness of soil. Do the hosts move around much in the olfactometers? The fact that they were contained with a mesh barrier suggests as much. If the antibiotic treatment suppressed host activity, then it would simultaneously affect both chemical and vibratory cues. Even if chemical cues are the primary modality used by the parasitoids, multimodal information should be considered as well.

L358-361: The proposed selective action of the antibiotic seems unlikely considering the previously stated widespread effectiveness of the same chemical (L303-305). The anatomy-based explanation for the results is still not compelling.

Author Response

Dear reviewer,

We are sending you the responses to your comments on our work.

We have included the changes in the manuscript with the tool “change control” and indicate the line numbers of the modifications in the new version. We wish to deeply thank you for the comments as we feel they have improved the article.

Follows a list the suggestions and how they have been considered throughout the text:

Reviewer: 1

L52-54: Please clarify “direct activity”, as I am not familiar with that term. The examples given don’t seem to fit together, as only plant chemicals are an example that doesn’t fit into the categories of originated from the hosts themselves or from associated microorganisms. Salivary secretions and host frass come from the host itself, right?

-We refer as direct activity as the normal activity or behaviours a host can have, like feeding or ovipositing on plants. Plant chemicals released in response to herbivory or oviposition are often cues for different parasitoids that use them. The examples given correspond to salivary secretions (host themselves), host frass (associated microorganisms), plant chemicals in response to herbivory or oviposition (direct activity). We included the words in response to herbivory and oviposition to clarify (lines 57-59 of the new version).

L62: Should this line specify “tachinid” species instead of “dipteran” species? As written, it is confusing

-We agree with the reviewer. We changed it as suggested (Line 68 of the new version)

L125: It was indicated earlier (L96-97) that numerous studies have already shown that larval symbionts are responsible for orientation cues used by M. ruficauda, so it doesn’t seem appropriate to include this species in part a) of the study goals.

-We agree with the reviewer. We removed the reference for M. ruficauda (line 142 of the new version). We had only included it in order to make comparable studies for the two species.

L131: Were larvae of both parasitoid species collected from all five sites? Documenting the degree to which these species overlap is important in understanding competition between them. The point above is especially important considering the results from the competition experiment. The extent of spatially,

temporal, or even micro-niche (e.g., soil depth or type) overlap between the species may preclude the evolution of meaningful responses to host infection status with another species?

-Yes. Both species coexist at the named sites. We have included a word (“both”, line 147 of the new version) that might help clarify. The only separation that may exist is the extent to which larvae can be wind dispersed. While M. ruficauda oviposits almost on the tip of the dry plants or wire fences, M. bigoti does it at a lower height. This might lower the competition in the field, but it has not been formally measured.

L147: Were soil samples taken near the apiaries?

-Yes. We included a clarification (lines 164-165 of the new version).

L157-162: The design of the olfactometer is important to understanding the methods and results (e.g., understanding how far the larvae must travel to find a host). Please include a figure of the device as a diagram and/or picture, especially since the citations provided do not include such a figure. Please clarify some details about the olfactometer. What is the box made of, plastic? Was soil present in the box during testing? How was the filter paper used, as a substrate in the box? Were the host and parasitoid placed into the olfactometer simultaneously?

-We have included a diagram of the experimental device (lines 203-211 of the new version). The box was made of plastic. No soil was present in the box, so no indication of it was included. The filter paper was placed between the box cover and the box, hence not interfering with the experiment. Some details have been included in lines 175 and 185-187 of the new version.

L172: If the parasitoids were only checked once after 60min, can it be stated with confidence that those observed in the middle zone had not left that zone? Isn’t it possible they had left the zone and returned at the time of data collection?

-We changed the word “remained” for “was found” (lines 191-192 of the new version).

L179-183: Unless there is a difference I don’t understand, this repeats information from L164-166.

-We do not believe they repeat information. On lines 164-166 it reads “Each individual chamber was divided into three equal size zones (one middle and two laterals) along the long axis (see [25,28,33]). As a stimulus, we used live hosts that were placed on one of the lateral zones and a plastic mesh that allowed the parasitoid larva to move freely around the arena but kept the host on the lateral zone.” while on lines 179-183 it reads “After every trial, each individual was removed and the arena was cleaned up with soap, water, ethyl alcohol and then dried with an air current in order to eliminate any possible remaining cue.”.

L188: How was the antibiotic applied to the larvae?

-We have included details (lines 215-218 of the new version). Now it reads “To this, we treated soil with tetracycline at 0.1 %w/w concentration and use it as the feeding substrate for host larvae of the most abundant species that were found at the collection sites of parasitoid larvae for seven days in the black tubes (30 ml) with pieces of carrot.”

Table 1: If “none” means that the host was unmanipulated, please clarify. I think a better term can be used to distinguish hosts without antibiotic treatment from the control runs in the olfactometer where there was no host present. These host-absent trials could also be called “none,” so there is potential for confusion here. Furthermore, the number of control series (with no host present) should be indicated in Table 1 or elsewhere in the methods. Were any individuals (parasitoids or hosts) used more than once? If not, please specify that important methodological detail. If so, please detail the extent of this and how any resulting pseudo-replication was handled.

-We changed “none” for “untreated” to distinguish hosts without antibiotic treatment and added the number of control series in Table 1. We also included that each individual (host and parasitoid) was used only once throughout the experiments (lines 198-199 in the new version). We also indicated the number of control series performed in Table 1.

L193-195: How much frass was used? Were the amounts similar between the two species tested in the frass assay? If so, that is interesting! If not, differences in the amount of frass could explain the differences in orientation behavior, obviating later

discussion.

-Indeed, amount of frass was similar between the species tested. We have weighted frass and checked that the amounts were similar. We included this information in lines 223-224 of the new version.

L204-206: How long ago were the hosts parasitized before use in this assay?

-We included this information in the manuscript (lines 235-238 of the new version). Hosts were parasitised between 7 and 10 days before because we have already shown that less time is not sufficient to change the chemical identity of the host.

L236: Should read “discard any bias,” though I think “assess” would be a more appropriate term than “discard.”

-We changed as suggested (line 266 of the new version).

Was the order of testing (for both hosts and parasitoids) controlled? Were randomized trials conducted, or might there by some systematic bias from testing one group before another (either over the whole course of the experiment or within each day of testing)?.

-Yes, indeed the order of testing was controlled taking care that at least one replicate within each experiment was done on the same day, although the order was randomized. We could include if needed more information on this aspect.

Egg cluster was considered as a random effect, but how was collection site handled? Furthermore, were hosts and parasitoids always paired from the same site, or were some individuals paired with hosts from a different locality? Fine-scale local adaptation to regional hosts (even within a species) could explain some of the results obtained.

-In effect, hosts and parasitoids were not always paired from the same site. Fine-scale local adaptation to regional hosts is a next step in understanding the process. However, the kind of experiments performed should expose responses where the survival of the individual is involved. At least in M. ruficauda, if a larva does not attach to a host as a second instar it cannot moult to the third instar. We have already shown that first instar larvae are able to moult to the second instar in absence of hosts. However, we do agree with the reviewer that this fine tuning is interesting to explore and will do in the future.

RESULTS

Please report the number (or percentage) of trials excluded because the parasitoid was found in the middle zone.

-We have included the percentage and the number of replicates of trials excluded in lines 277-280.

L252-253: The conclusion that the orientation response was lost upon antibiotic treatment is an interpretation of the data, better suited for the discussion section.

-We rephrased the results in lines 284-287 of the new version.

L253-255: It’s unclear what is meant by “orientation patterns.” For CS and CP, there was orientation towards the untreated host, but no orientation towards the untreated host. However, CS and CP were not similar in terms of the effectiveness of antibiotic

in mediating the orientation response, as only the odds ratio for CS was significant. This point should be made more clearly.

-This point has been addressed in the discussion regarding the marginal significance of CP. We sustain that there is a source of variability that does not allowed us to show so clearly the effect, but it is undeniable that there seems to be an effect. See more on this below in the answer of lines 302-303.

L268-269: As above, this needs to be phrased to better match the statistics. The only significant orientation response was to CS, based on the odds ratios. Because of this, both parasitoid species have comparable responses: significant orientation towards CS but no orientation towards other (not shared) species.

-We apologize for an involuntary mistake we have done on the odds ratio for CS. Odds ratio for CS are very similar to HB based also on the estimated probabilities or each group (we corrected the UCI for the CS group in Table 3). However, an important note should be made. In this case, odds ratio shows no difference between the treated and untreated groups for CS, HB and PB. However, the estimated probabilities of orientation in CS and HB are highly significant (>.75, Table 3) while the estimated probabilities for PB are not significant. This means that the odds ratio is showing no difference within treated groups but combined with the estimated probabilities we can conclude that positive orientation to CS and HB is found in M. bigoti.

L271: Save the interpretation of the data (i.e., use of different orientation cues) for the discussion.

-We have deleted the interpretation (lines 305-306 of the new version).

Table 3. What are the 2nd/3rd indicators in the table? The methods (L149-151) indicated only third instar hosts and second instar parasitoids were used. I expected the table to be formatted with merged cells for each host, just as was done for CM, HB, and PB.

-We apologize for that mistake; it was deleted as it does not correspond to this study.

Table 3. The UCI for PB has two decimal points and is missing the closed parenthesis.

-We have corrected it.

Figure 2. The bar graph display is not very informative for these data. This would be more effectively displayed as a box plot with a reference line at 0.5 probability. Then, confidence intervals overlapping 0.5 would indicate no orientation response in that

treatment. Confidence intervals below 0.5 would indicate avoidance, above would indicate attraction.

-We do not agree with the reviewer. Box plot is commonly used as a descriptive technique for raw data. In fact, IQR (at least by default) is calculated assuming a normal distribution that is not applicable here. However, we acknowledge that the information from the figure is already included in Table 3, so we decided to remove the graph.

L282: The reference should be for Table 4. Table 4 formatting is odd, raising the same questions I had about Table 3.

-We changed it as suggested (line 316 of the new version).

DISCUSSION

L302-303: According to the odds ratio, antibiotic did not affect the orientation to CP.

-Although we do concur with the reviewer that the odds ratio at .05 level does not support our claim, we believe that this is just a statistical issue. In fact, CP comparison is made between .827 vs .525 while CS comparison is made between .871 vs .505 (Table 3). Given the number of replicates of the experiments if only 1 replicate would have been different the significance at .05 level would have been found. Moreover, if instead of using .05 level, we would have used .10 this result would have been stated as statistically different. What we intend to expose here is that we do not agree that there is no effect of the antibiotic, and the difference might probably be because of variability not controlled. Despite this, it is evident that the effect was not so strong as that found for CS. We have included some disclaimer on this point in the new version for discussion (lines 336-340 of the new version).

L316-317: Is there any evidence of antibiotic having location-specific effects? It seems far more likely that cues of another modality are used, though the frass results are intriguing. Have tests been done to test the efficacy of the antibiotic? In cases where there do not appear to be antibiotic effects, it could be that the initial dose was not sufficient to clear the symbionts from the host. Alternatively, re-infection from the environment (e.g., carrots or soil) could help explain these results.

-No, tests on efficacy of the antibiotic have been performed because we had already tested this way and amount of dosage in previous works on C. signaticollis. Although it can be true that the initial dose was not sufficient, we do not believe that reinfection was possible since soil was treated with tetracycline, so the only soil they ingested during that time was the one with tetracycline. Pieces of carrots were also in contact with the soil, so they were also in contact with the antibiotic. However, we concur that more experiments with different doses and ways of administering the tetracycline are needed. Another possibility could be that tetracycline is not so effective in the colon where frass is produced leaving host frass unaltered when treated with tetracycline, or perhaps, the microorganismal community from the colon is very different from that of the hindgut and more resistant to tetracycline treatment. We have developed these ideas in lines 355-360 of the new version.

L321-322: Given the uncertainties about antibiotic efficacy, it seems premature to conclude anything about the different parts of the host creating cues. This conclusion reaches beyond the data and seems less plausible than other explanations. The arguments about parasitoid density (L331-344) are compelling, but another highly plausible alternative should be considered: the parasitoid may still orient toward an infected host, but it may then reject that host after detecting more information. The studies presented here only represent an early step in the parasitism process: locating a host. After location, the parasitoid would have further opportunities to reject the previously infected host and seek another.

-We agree with the reviewer. However, we have done acceptance experiments where larvae of M. bigoti were offered C. signaticollis hosts. Our results show that untreated hosts are highly parasitised (~83%, N=94) but treated hosts are rejected (~50%, N=20). We did not include acceptance results because we only have partial results on some of the species. Although these results are obtained from healthy hosts, they seem to support that C. signaticollis is an accepted host. Combined with the hypothesis that the chemical identity of the host changes after several days since parasitism, then multiparasitism is a possible scenario. We intend to explore how competition is resolved under multiparasitism. We included some information on this on lines 366-368 of the new version.

L345-346: The argument for a reliance on chemical cues is compelling, but not complete. Have vibratory cues from the hosts been ruled out as a potential signal for seeking parasitoids? Vibrations are an important source of information in the darkness of soil. Do the hosts move around much in the olfactometers? The fact that they were contained with a mesh barrier suggests as much. If the antibiotic treatment suppressed host activity, then it would simultaneously affect both chemical and vibratory cues. Even if chemical cues are the primary modality used by the parasitoids, multimodal information should be considered as well.

-We sustain that vibratory cues are not useful under our experimental device. The arena had no soil, and the hosts were placed on the plastic. This material has little possibility of transmitting vibratory signals. We are not claiming that they are not involved, but at least in our setup they would not be important. Hosts can move freely although in a limited space. Activity is not suppressed; they are readily active and show normal behaviour after antibiotic treatment.

L358-361: The proposed selective action of the antibiotic seems unlikely considering the previously stated widespread effectiveness of the same chemical (L303-305). The anatomy-based explanation for the results is still not compelling.

-We sustain that tetracycline is an antibiotic for many different microorganisms but that it has different effectiveness on different host parts. Hence, it may be more effective in the fermentation chamber than on any other part and still attack many different microorganisms.

-We thank you for the suggestions that really improved the manuscript. Thanks for a thoughtful and constructive review.

Reviewer 2 Report

Title: The title is misleading because the results do not show any involvement of Bacteria. this could therefore be improved. 

Abstract: Lines 16-17, 19-24, and 25-26 could be improved to clearly document the objective and results of the study.

Introduction: line 34, could read "...insect could be considered a multi-organism..."

Line 53, could be improved

Section 2.2. on Experimental Procedures, consider to use a drawing. the description in the text is hard to follow especially on the design and replications.

Section 2.3 it is not clear how the antibiotic was administered

Table 1. Scientific names need be italicized

Results:

Lines 246-248: different orientations of the arena are tested for differences. results should therefore be given for the two species of insects as the tests must have been performed separately.

Lines 252 - 255: The sentences do not reflect the results in Table 2 and Figure 2.

Table 3 could be presented differently to enhance clarity.

Figure 2: there are no statistical differences between treated and no treatment, and between species. the bars are 95% CI.

Lines 269-271 contradict preceding sentences

Line 282: should be Table 4.

Table 4 could be presented differently to enhance clarity

Discussion

Line 302-303: results do not show loss of attraction

Lines 311-317: the lines need correction. the results presented do not reflect that text.

Line 345 could be improved in consideration of presented results.

Line 365-366: the results presented here do not show involvement of chemical cues nor effect of microorganisms.

Author Response

Dear reviewer,

We are sending you the responses to your comments on our work.

We have included the changes in the manuscript with the tool “change control” and indicate the line numbers of the modifications in the new version. We wish to deeply thank you for the comments as we feel they have improved the article.

Follows a list the suggestions and how they have been considered throughout the text:

Reviewer: 2

-Title: The title is misleading because the results do not show any involvement of Bacteria. this could therefore be improved.

-Done. We changed the title for microorganisms, since we sustain that our results with antibiotics show sufficient evidence for a microorganismal role in cue production.

-Abstract: Lines 16-17, 19-24, and 25-26 could be improved to clearly document the objective and results of the study.

-In line with the previous response, we changed were necessary with “microorganism”.

-Introduction: line 34, could read "...insect could be considered a multi-organism..."

-We changed as suggested (line 39 of the new version).

-Line 53, could be improved

-We have changed it (lines 57-59 of the new version).

-Section 2.2. on Experimental Procedures, consider to use a drawing. the description in the text is hard to follow especially on the design and replications.

-Done. We added a figure with a drawing of the experimental arena, and the possible parasitoid responses according to the decision areas (lines 203-211 of the new version).

-Section 2.3 it is not clear how the antibiotic was administered

-We have included more information (lines 215-218 of the new version).

-Table 1. Scientific names need be italicized

-We have changed them.

Results:

-Lines 246-248: different orientations of the arena are tested for differences. results should therefore be given for the two species of insects as the tests must have been performed separately.

-We respectfully disagree with the reviewer. The control series were performed for each species separately but the analyses we performed was a Pearson's Chi-squared test with Yates' continuity correction. We tested if orientation to one side of the arena was different from a random orientation in both species. Since it is only a two category analyses should the test had thrown significant differences, we would have known to which species was corresponding by looking at the proportion of orientation to each side.

-Lines 252 - 255: The sentences do not reflect the results in Table 2 and Figure 2.

Table 3 could be presented differently to enhance clarity.

-Tables show estimated probabilities for orientation compared to 0.5. We assume the reviewer is referring to Table 3 since it is the one showing results. In Table 3 the probability column shows these estimated probabilities with all the information and the p value corresponding to the significance against 0.5 null hypothesis. In that table you can see that orientation to untreated CP and CS is significant. The OR (odds ratio) column is showing the comparison for each host species between treated and untreated groups. In this case, the odds ratio would be significant if the number 1 is not included in the confidence interval of the odds ratio. Given this explanation we sustain the results of the Table 3. Figure 2 had been removed after following indication of reviewer 1 and noticing that it already contained the same information as Table 3.

-Figure 2: there are no statistical differences between treated and no treatment, and between species. the bars are 95% CI.

-It has been removed.

-Lines 269-271 contradict preceding sentences

-We respectfully disagree with the reviewer. Preceding sentences are referring to the other parasitoid species M. ruficauda that shows orientation to different hosts species.

-Line 282: should be Table 4.

-We have changed it.

-Table 4 could be presented differently to enhance clarity

-We hope that with the explanation given for Table 3 is now clearer.

Discussion

-Line 302-303: results do not show loss of attraction.

-These results are based on the analysis of the odds ratio. The odds ratio between groups compares estimated probabilities and calculates new confidence intervals. If both groups were the same number, then the odds ratio would have given 1. This is why, if the confidence interval for the odds ratio does not contain the number 1 shows significant differences between the compared groups.

-Lines 311-317: the lines need correction. the results presented do not reflect that text.

-These lines are also interpreted with the explanation given before and the analysis of the odds ratio. We hope that now the analysis is clearer.

-Line 345 could be improved in consideration of presented results.

-We do not understand how it can be improved based on the presented results.

-Line 365-366: the results presented here do not show involvement of chemical cues nor effect of microorganisms.

-We have shown for M. ruficauda the involvement of chemical cues through different methodologies and performing even similar experiments like the ones presented here but with host extracts as odour sources. Here, we are presenting live hosts as stimuli that is a more complete cue. Indeed, vibratory cues could be acting, but as discussed with the other reviewer, these cues cannot reach far in a plastic box thus we discard involvement of this modality. Also, treatment with tetracycline is accepted as a generalist antibiotic that attacks many different microorganisms including bacteria. And given that hosts belong to the Scarabaeidae family that have symbionts in the fermentation chamber we believe it is rather well demonstrated the involvement of chemical cues and microorganisms.

-We thank your for the suggestions that improved the manuscript.